# Self Iterative Label Refinement
# via Robust Unlabeled Learning

**Hikaru Asano**
The University of Tokyo
Tokyo, Japan
asano-hikaru19@g.ecc.u-tokyo.ac.jp

**Tadashi Kozuno**
OMRON SINIC X
Tokyo, Japan
tadashi.kozuno@sinicx.com

**Yukino Baba**
The University of Tokyo
Tokyo, Japan
yukino-baba@g.ecc.u-tokyo.ac.jp

## Abstract

Recent advances in large language models (LLMs) have yielded impressive performance on various tasks, yet they often depend on high-quality feedback that can be costly. Self-refinement methods attempt to leverage LLMs' internal evaluation mechanisms with minimal human supervision; however, these approaches frequently suffer from inherent biases and overconfidence, especially in domains where the models lack sufficient internal knowledge, resulting in performance degradation. As an initial step toward enhancing self-refinement for broader applications, we introduce an iterative refinement pipeline that employs the Unlabeled-Unlabeled learning framework to improve LLM-generated pseudo-labels for classification tasks. By exploiting two unlabeled datasets with differing positive class ratios, our approach iteratively denoises and refines the initial pseudo-labels, thereby mitigating the adverse effects of internal biases with minimal human supervision. Evaluations on diverse datasets, including low-resource language corpora, patent classifications, and protein structure categorizations, demonstrate that our method consistently outperforms both initial LLM's classification performance and the self-refinement approaches by cutting-edge models (e.g., GPT-4o and DeepSeek-R1). Moreover, we experimentally confirm that our refined classifier facilitates effective post-training alignment for safety in LLMs and demonstrate successful self-refinement in generative tasks as well.[1]

## 1 Introduction

Rapid advancements in large language models (LLMs) have yielded significant improvements across various downstream tasks and have raised a fundamental research question: *How can we further improve an LLM's capabilities with minimal human supervision?* Traditional approaches, such as Reinforcement Learning from Human Feedback (RLHF) [40] and its variants [42, 53], improve performance through classification tasks [53], yet they rely on extensive, costly, and time-consuming human-labeled datasets. Although recent methods like Reinforcement Learning from AI Feedback (RLAIF) [3, 29, 67] aim to reduce these costs by replacing human annotations with model-generated signals, their success critically hinges on the reliability of the model's self-evaluation [30, 58].

---

[1]Our code is available at https://github.com/HikaruAsano/self-iterative-label-refinement.

39th Conference on Neural Information Processing Systems (NeurIPS 2025).

When using an LLM as its own evaluator, it is observed that the model's inherent biases can harm the reliability of its assessments [24, 31, 34, 58], thereby undermining the effectiveness of downstream training [4]. While iterative self-refinement [23, 37] and multi-agent frameworks [16] can mitigate simpler biases (e.g., ordering or length biases), LLMs still encounter significant challenges in areas where their internal knowledge is limited [21]. In such cases, external tools, such as accurate, domain-specific databases, can help address certain blind spots [61], but they do not fully eliminate the need for human supervision. Without either robust internal knowledge or dependable huge human input, conventional methods not only struggle to improve performance [18, 21] but also may even experience degradation due to inaccuracies and overconfidence [18].

To address these challenges specifically for classification tasks, a crucial first step towards broader self-refinement in LLMs, we introduce an iterative pipeline that refines LLM-generated pseudo-labels using a weakly supervised learning technique known as the Unlabeled-Unlabeled (UU) learning framework [35, 36]. Notably, while creating a large, well-maintained labeled dataset requires intensive human supervision, it is relatively straightforward to amass a vast corpus of unlabeled data in the modern era [45]. Motivated by this observation, our approach leverages two unlabeled datasets with differing positive class ratios. Under the simple assumption that one dataset contains a higher proportion of positive examples than the other, our system can effectively learn to distinguish between positive and negative instances without requiring explicit annotations for each example.

Specifically, our pipeline first employs an LLM to generate initial pseudo-labels from an unlabeled corpus. These pseudo-positive and negative sets are then iteratively refined using UU learning. The classifier trained from UU learning subsequently re-labels the unlabeled corpus, progressively reducing noise and enhancing classification accuracy. By decoupling the refinement process from the LLM's internal knowledge and instead leveraging data-driven features extracted via UU learning, our method delivers improved performance even in domains where LLMs lack sufficient knowledge.

We evaluate our approach on several public datasets, including low-resource language corpora, patent classification tasks, and protein structure classification. Notably, as illustrated in Figure 3, even in cases where self-refinement methods based on LLMs, or advanced reasoning models such as DeepSeek-R1 [13], fail to produce any performance improvement, *our iterative UU learning framework successfully refines its outputs and achieves classification performance that surpasses that of both the original LLM and existing self-improvement pipelines*. Moreover, we experimentally show that our refined classifiers, integrated into RLAIF frameworks, effectively *achieve safety alignment without extensive human annotation*. This underscores our method's potential for comprehensive, robust LLM self-refinement.

In summary, our contributions are threefold: (i) We introduce an iterative pipeline that refines LLM-generated pseudo-labels via UU Learning, reducing noise and boosting classification accuracy with minimal human supervision; (ii) we demonstrate our method consistently surpasses direct LLM classification and existing self-improvement methods across diverse tasks (e.g., low-resource languages, patents, proteins), enabling scalable, high-quality classification with limited labeled data; and (iii) we experimentally show that our refined classifier facilitates effective post-training alignment for LLMs and highlight its potential for self-refinement in broader generative tasks.

## 2   Related Work

RLAIF is a popular LLM post-training method [3]. Its idea to generate feedback by a model itself is called Pseudo-Labeling (PL) and well studied in semi-supervised learning [28, 48]. Below, we will review PL and LLM self-training methods related to ours. For a thorough review of each topic, please refer to Yang et al. [63] and Xiao and Zhu [62].

**Pseudo-Labeling:**   PL trains a student model so that its output is close to the output of a teacher model on unlabeled data [28, 48]. Various methods for constructing a teacher model has been proposed. For example, $\Pi$-model [2, 27, 46] and virtual adversarial training [39] perturbs input and/or neural networks of student models. Some methods use weighted average of previous student models' predictions or weights as a teacher model [27, 55], and other methods even try to optimize a teacher model by viewing PL as an optimization problem [41, 57, 65]. Our work is complimentary to this line of works since what we modify is the risk estimator rather than teacher models.

PL is known to suffer from erroneous labels generated by a model-in-training [17, 43], and this observation well aligns with recent reports on RLAIF that erroneous self-feedback is a major source of failure [37]. A straightforward approach is to filter potentially incorrect labels based on confidence [9, 17, 20, 43, 52] or the amplitude of loss as in self-paced learning [6, 20, 25]. Another method is refining pseudo-labels in a way similar to label propagation [68] but with similarity scores computed using neural networks [26]. Our work tackles the issue of erroneous labels by using a risk estimate robust to erroneous self-feedback based on UU learning [35, 36]. Even though our approach requires only a minimal change, we observed a significant performance boost.

**LLM's Self-Refinement:** To enhance the reasoning capabilities of LLMs, early efforts primarily explored various prompt engineering techniques [8, 59, 60, 64]. Even with refined prompting strategies, an LLM's initial outputs can remain limited in some scenarios. Recent studies have proposed iteratively improved answer strategies called self-refinement approaches [11, 23, 37], and our work falls within this lineage.

Self-refinement involves generating responses through agents in distinct roles that iteratively provide feedback [50, 69]. For example, multi-agent debate frameworks [16] use an answering agent and an evaluation agent to collaboratively improve answer quality [10, 15, 51]. However, these methods usually assume that the LLM has enough internal knowledge to generate effective feedback and revisions; when it doesn't, performance gains can be minimal or even negative [18, 21, 22, 32], sometimes degrading performance [18]. Our approach minimizes this reliance: internal knowledge is used only initially for classification, with subsequent improvements relying on extracting features directly from the data via UU learning.

Other work addresses knowledge gaps by retrieving external data or using external tools [19, 49, 56, 61], though such setups can be costly. In contrast, our method needs only a small amount of labeled data for initialization, without relying on external resources.

## 3 Preliminaries

### 3.1 Supervised Binary Classification

In many real-world tasks, one commonly encounters binary classification problems, in which an input $x \in \mathbb{R}^d$ is presented, and its label $y \in \{\pm 1\}$ needs to be predicted. Each sample is assumed to be independently and identically drawn from an unknown joint distribution $p(x, y)$. Let $\pi_+ = p(y = +1)$ be the prior probability of the positive class (positive prior), and define

$$p_{\mathrm{p}}(x) = p(x \mid y = +1), p_{\mathrm{n}}(x) = p(x \mid y = -1).$$

Then, the marginal distribution of $x$ is given by

$$p(x) = \pi_+ p_{\mathrm{p}}(x) + (1 - \pi_+) p_{\mathrm{n}}(x).$$

A classifier $g : \mathbb{R}^d \to \mathbb{R}$ outputs a real-valued score, whose sign determines the predicted label. For instance, a neural network can serve as $g$. A loss function $\ell : \mathbb{R} \times \{\pm 1\} \to [0, \infty)$ then measures how much the prediction disagrees with the true label. Let $R_p^+(g) = \mathbb{E}_{x \sim p_p}[\ell(g(x), +1)]$ denote the loss for true positive data, and $R_n^-(g) = \mathbb{E}_{x \sim p_n}[\ell(g(x), -1)]$ denote the loss for the true negative data. Then, the true risk is expressed as

$$
\begin{aligned}
R_{\mathrm{pn}}(g) &= \mathbb{E}_{(x,y) \sim p}[\ell(g(x), y)] \\
&= \pi_+ R_p^+ + (1 - \pi_+) R_n^-
\end{aligned}
\tag{1}
$$

In supervised learning, positive dataset $\mathcal{C}_p = \{x_m^p\}_{m=1}^{m_p} \sim p_p(x)$ and negative dataset $\mathcal{C}_n = \{x_m^n\}_{m=1}^{m_n} \sim p_n(x)$ are accessible. Replacing the expectations in (1) with sample mean, one obtains the empirical risk, and $g$ is trained to minimize it.

It is well known that having sufficient positive and negative samples typically allows one to train a highly accurate classifier for many tasks. However, in practice, obtaining large-scale positive and negative datasets with annotations is often challenging, especially in specialized domains where annotation costs become a significant obstacle.

## 3.2 Unlabeled-Unlabeled (UU) Learning

UU learning [35] is a technique that allows training a classifier without fully labeled positive and negative datasets, leveraging two unlabeled datasets with different class priors.

Concretely, suppose unlabeled corpora, $\widetilde{\mathcal{C}}_p = \{\widetilde{x}_m^p\}_{m=1}^{m_p}$ and $\widetilde{\mathcal{C}}_n = \{\widetilde{x}_m^n\}_{m=1}^{m_n}$, drawn from different mixture distributions. We denote $\theta_p = p(y = +1 \mid \widetilde{x} \in \widetilde{\mathcal{C}}_p)$ and $\theta_n = p(y = +1 \mid \widetilde{x} \in \widetilde{\mathcal{C}}_n)$ the *positive prior* of these unlabeled corpora. In other words, $\theta_p$ is the fraction of true positives in $\widetilde{\mathcal{C}}_p$, and $\theta_n$ is the fraction of true positives in $\widetilde{\mathcal{C}}_n$. Then, the mixture distribution of each corpus is given as

$$\widetilde{p}_p(x) = \theta_p\, p_p(x) + \big(1 - \theta_p\big)\, p_n(x), \quad \widetilde{p}_n(x) = \theta_n\, p_p(x) + \big(1 - \theta_n\big)\, p_n(x).$$

When $\theta_p > \theta_n$, we can treat $\widetilde{\mathcal{C}}_p$ as a pseudo-positive corpus (due to its larger proportion of actual positives) and $\widetilde{\mathcal{C}}_n$ as a pseudo-negative corpus (having a smaller proportion of actual positives).

By appropriately combining these two unlabeled sets, one can construct an unbiased estimate of the true binary classification risk (1). Specifically, let $R_{\widetilde{p}}^{\pm}(g) = \mathbb{E}_{x \sim \widetilde{p}_p}[\ell(g(x), \pm 1)]$, and $R_{\widetilde{n}}^{\pm}(g) = \mathbb{E}_{x \sim \widetilde{p}_n}[\ell(g(x), \pm 1)]$. Then, the UU learning risk is given by

$$R_{\mathrm{uu}}(g) = a R_{\widetilde{p}}^+(g) - b R_{\widetilde{p}}^-(g) - c R_{\widetilde{n}}^+(g) + d R_{\widetilde{n}}^-(g), \tag{2}$$

where the coefficients $a$, $b$, $c$, $d$ are computed from $\pi_+$, $\theta_p$, and $\theta_n$ as $a = \frac{(1-\theta_n)\,\pi_+}{\theta_p - \theta_n}$, $b = \frac{\theta_n\,(1-\pi_+)}{\theta_p - \theta_n}$, $c = \frac{(1-\theta_p)\,\pi_+}{\theta_p - \theta_n}$, $d = \frac{\theta_p\,(1-\pi_+)}{\theta_p - \theta_n}$. When $\theta_p = 1$ and $\theta_n = 0$, that is, when using the same dataset as standard supervised learning, equation (2) reduces to the standard supervised learning risk equation (1). In other words, supervised learning can be considered a special case of UU learning.

## 3.3 Robust UU Learning

While UU learning (2) does allow model training without explicit positive/negative labels, comparing the original binary classification risk (1), which remains nonnegative, against the UU risk (2) shows the UU risk includes negative terms such as $-b R_{\widetilde{p}}^-(g)$ and $-c R_{\widetilde{n}}^+(g)$. It has been observed that these negative risk terms can lead to overfitting [36].

To mitigate this, *Robust UU Learning* introduces a generalized Leaky ReLU function $f$ to moderate the reduction of negative risk. Concretely, it normalizes each term of the loss function as [36]

$$R_{\mathrm{ruu}}(g) = f\left(a R_{\widetilde{p}}^+(g) - c R_{\widetilde{n}}^+(g)\right) + f\left(d R_{\widetilde{n}}^-(g) - b R_{\widetilde{p}}^-(g)\right) \tag{3}$$

where each bracketed term resembles a "normalized" risk under the hypothetical label of being positive or negative, respectively. The function $f$ is defined as $f(x) = x$ if $x > 0$, and $f(x) = \lambda x$ if $x < 0$, where $\lambda < 0$. Intuitively, $f$ preserves positive risk values while converting negative risk into positive values using $\lambda < 0$, thereby reducing overfitting caused by negative risk terms.

# 4 Method

Below, we present our iterative framework for combining LLM annotations with *robust UU learning*. The goal is to iteratively refine pseudo-labels generated by the LLM, thus boosting classification accuracy under minimal human supervision. Figure 1 provides an overview of this pipeline.

## 4.1 Overview of the Iterative Pipeline

Our pipeline proceeds in three steps, (i) **LLM-Based Annotation (Iteration 0)**: an LLM provides pseudo-labels for an unlabeled corpus (§4.2); (ii) **Robust UU Learning**: we split the corpus into *pseudo-positive* and *pseudo-negative* subsets and train a classifier via robust UU learning (§4.3); and (iii) **Re-Labeling**: the trained classifier re-labels the entire dataset, producing refreshed pseudo-positive and pseudo-negative sets for the next iteration (§4.4).

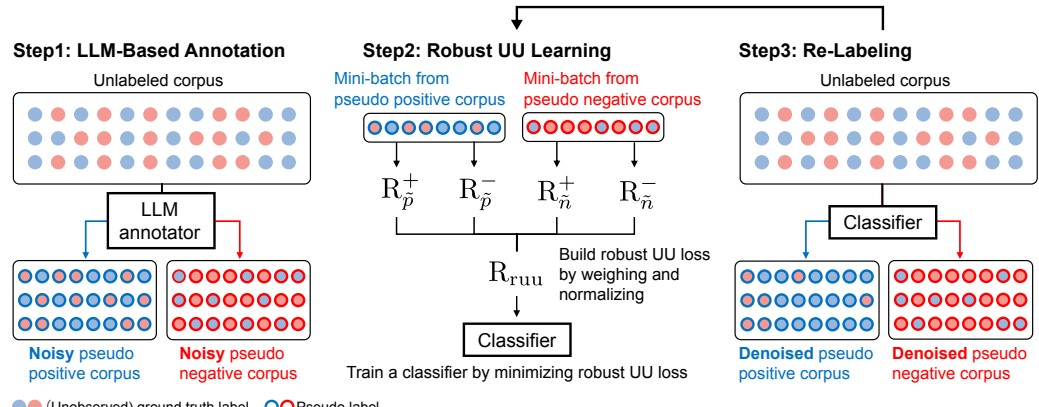

Figure 1: Overview of our iterative refinement pipeline. First, an LLM annotator generates initial pseudo-labels for an unlabeled corpus, dividing it into pseudo-positive and pseudo-negative corpora. Next, we train a classifier using robust UU learning on these pseudo corpora, yielding a model that outperforms the initial LLM annotations. Finally, the classifier re-labels the entire dataset, updating the pseudo-labels for the next iteration. Repeating this cycle gradually refines the pseudo-labels, leading to increasingly reliable labels.

## 4.2 Initial Noisy Annotation via LLM

Let $\mathcal{C} = \{x_1, x_2, \ldots, x_N\}$ be a corpus of unlabeled samples. We use the LLM as the initial classifier to assign an initial pseudo-label $\tilde{y}_i \in \{+1, -1\}$ to each sample $x_i$. Our prompt first provides a concise description of the task, the dataset domain, and the expected answer format (e.g., "Output TRUE or FALSE"). We then give a few-shot examples illustrating how to label an example text, along with a short rationale. Finally, the prompt includes the samples to annotate (see Figure 5 for the exact prompt).

Based on the LLM's output, we form two unlabeled corpora:

$$\widetilde{\mathcal{C}}_p^{(0)} = \{\, x_i \mid \tilde{y}_i = +1\}, \widetilde{\mathcal{C}}_n^{(0)} = \{\, x_i \mid \tilde{y}_i = -1\}.$$

These sets are called *pseudo-positive* and *pseudo-negative* corpus, respectively. Although the labels are noisy, $\widetilde{\mathcal{C}}_p^{(0)}$ typically has a higher positive prior than $\widetilde{\mathcal{C}}_n^{(0)}$, thereby providing a reliable foundation for UU learning in subsequent iterations.

## 4.3 Refinement with Robust UU Learning

Let $\widetilde{\mathcal{C}}_p^{(t-1)}$ and $\widetilde{\mathcal{C}}_n^{(t-1)}$ denote the pseudo-positive and pseudo-negative sets from iteration $t-1$. Our goal is to train a classifier $g^{(t)}$ (e.g., a neural network) despite noisy labels. To this end, we optimize the *robust UU learning* objective:

$$g^{(t)} \;=\; \operatorname{argmin}_{g \in \mathcal{G}} \widehat{R}_{\mathrm{ruu}}\big(g;\, \widetilde{\mathcal{C}}_p^{(t-1)},\, \widetilde{\mathcal{C}}_n^{(t-1)}\big),$$

where $\widehat{R}_{\mathrm{ruu}}(\cdot)$ is the empirical risk of robust UU learning, which applies a "generalized leaky ReLU" to reduce the impact of negative risk terms that can arise from mislabels. Each set is weighted by the positive prior $\pi_+$ and the sets' own estimated positive priors $\hat{\theta}_p$ and $\hat{\theta}_n$.

This robust learning approach is less sensitive to initial label noise and can produce a classifier that outperforms the previous iteration's classifier.

## 4.4 Iterative Re-Labeling and Convergence

After training $g^{(t)}$, we re-label the entire dataset: $\tilde{y}_i^{(t)} := \operatorname{sign}\big(g^{(t)}(x_i)\big) \in \{+1, -1\}$. From these labels, we form updated *pseudo-positive* $\widetilde{\mathcal{C}}_p^{(t)} = \{\, x_i \mid \tilde{y}_i^{(t)} = +1\,\}$ and *pseudo-negative* $\widetilde{\mathcal{C}}_n^{(t)} = \{\, x_i \mid \tilde{y}_i^{(t)} = -1\,\}$ sets, used in the next robust UU learning iteration.

Over several iterations, this process progressively improves the reliability of the pseudo-labels. In the ideal scenario, the positive prior in $\widetilde{\mathcal{C}}_p^{(t)}$ converges to 1, and the positive prior in $\widetilde{\mathcal{C}}_n^{(t)}$ converges to 0, bringing each corpus ever closer to the gold-standard case of perfectly labeled positive and negative data. When these priors reach 1 and 0, respectively, robust UU learning effectively reduces to standard supervised learning, achieving high accuracy even from initially noisy labels.

# 5 Experiments

We conducted experiments to explore three main research questions:

**RQ1** Can our iterative refinement approach improve classification performance, compared to the initial LLM-based annotations, for various NLP tasks?

**RQ2** Can our approach enhance performance on challenging tasks where even advanced LLMs (e.g., GPT-4o or DeepSeek-R1) struggle?

**RQ3** Can our method extend beyond classification tasks to generative tasks such as LLM alignment?

## 5.1 Experimental Setup

**Datasets:** We use six binary classification datasets grouped into two categories based on their difficulty[2]. Table 2 reports the dataset statistics, and Table 3 provides examples for positive and negative cases (see Appendix B).

**Easier Tasks (for RQ1):** We evaluated our algorithm on three tasks: (i) **Fake News** [1], classifying news articles as fake or real, (ii) **Saroco** [44], Romanian satire detection dataset to assess effectiveness in a low-resource language, and (iii) **Safety** [12], dataset for SafeRLHF evaluating if the responses to questions is safe or dangerous, thereby assessing effectiveness for LLM's post-training.

**Harder Tasks (for RQ2):** We evaluate our algorithm on three more challenging tasks: (i) the **Corona Sentiment**[3], classifying social media post related to COVID-19 as positive or negative sentiment, (ii) the **Green Patent**[4] which involves identifying whether a patent abstract pertains to green plastics, requiring high expert knowledge, and (iii) **Protein Structure** [7], involving classification of proteins based on high or low COVID-19 binding affinity from molecular SMILES strings.

For all experiments, we randomly partitioned each dataset into training, validation (for best epoch selection), and test splits in a 7:1:2 ratio.

**LLM-Based Annotation (Iteration 0):** At *iteration0*, we obtain pseudo-labels from LLMs using a prompt that includes a dataset explanation, a few-shot labeled examples, and a target sample (see Figure 5 for the exact prompt). These pseudo-labels are then used to construct initial pseudo-positive ($\widetilde{\mathcal{C}}_p^{(0)}$) and pseudo-negative ($\widetilde{\mathcal{C}}_n^{(0)}$) corpora.

**Training Procedure:** We train the classifier by appending an affine layer to the transformer's final hidden state to yield a one-dimensional score. For fine-tuning efficiency, we employ QLora [14] with 4-bit quantization. At each iteration, we fine-tune the model using the pseudo-positive and pseudo-negative corpora with the AdamW optimizer (learning rate = $1.0 \times 10^{-4}$, batch size = 16, and 3 epochs) and fix the robust UU learning hyperparameter $\lambda$ at -0.001. At the end of each epoch, we compute the loss on a pseudo-labeled validation set and select the model with the lowest loss to re-label the entire dataset. Additional parameters are detailed in Table 4.

**Estimating Class Priors:** Robust UU learning requires estimates of the positive proportions $\hat{\theta}_p$ and $\hat{\theta}_n$ for the pseudo-positive and pseudo-negative corpora, respectively. We compare two settings to obtain these parameters. In the first setting **(Ours (Oracle))**, we use the exact values of $\theta_p$ and

---

[2]We evaluate difficulty based on a pilot experiment in which we trained the classification model in a standard supervised setting. Please refer to Table 2 for classification accuracy via supervised learning.

[3]`https://github.com/akshayjoshii/COVID19-Tweet-Sentiment-Analysis-and-EDA/tree/master`

[4]`https://huggingface.co/datasets/cwinkler/patents_green_plastics`

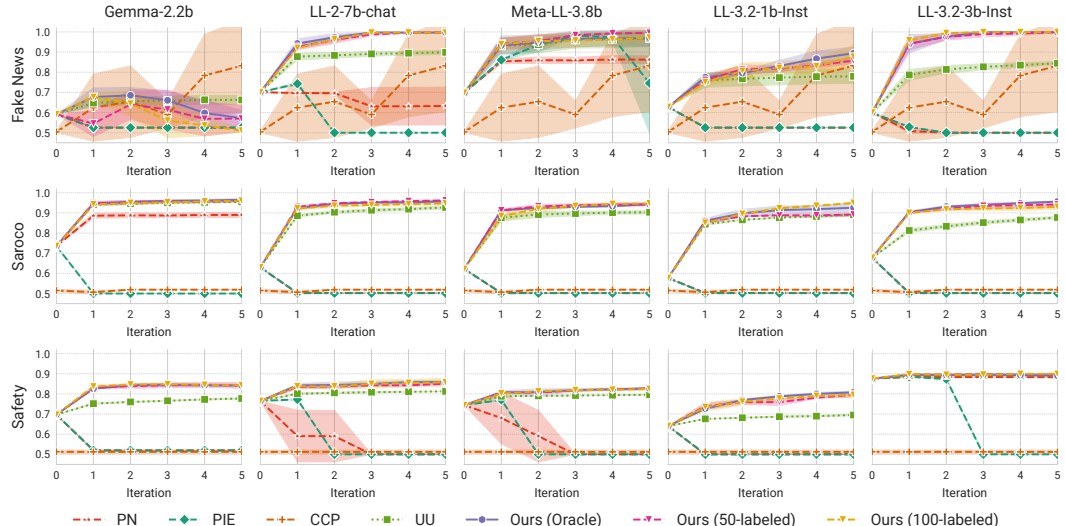

Figure 2: Classification accuracy over five iterations for three datasets. The solid lines represent the mean values, and the shaded areas show the mean $\pm$ standard deviation. Both variants of our approach, Ours (Oracle) and Ours (few-labeled), demonstrate steady improvements as the iteration increases. Notably, even in scenarios where the baselines fail to learn the classification task, our method continues to exhibit iterative performance gains. This robustness highlights the strength of our iterative refinement strategy, even under minimal supervision settings like 50 labeled examples. Detailed numerical results are provided in Appendix D.

$\theta_n$, which represents a *theoretical upper bound* on performance[5]. In the second setting (**Ours (few-labeled)**), we assume a practical scenario with only a small subset of labeled data (50 or 100 examples). From these examples, we estimate the class priors by computing $\hat{\theta}_p = p(y = 1 \mid \tilde{y} = 1)$ and $\hat{\theta}_n = p(y = 1 \mid \tilde{y} = 0)$ before applying robust UU learning. Our experiments show that even a small amount of labeled data provides sufficiently accurate estimates.

**Evaluation Metrics and Protocol:** We measure classification accuracy on the held-out test dataset across all iterations. Each run is repeated with three different random seeds, and we report the mean and standard deviation.

## 5.2 Experiments on Easier NLP Tasks (RQ1)

We start with three relatively simple tasks: **Fake News**, **Saroco**, and **Safety**. For the base model of the classifier, we use Llama-3.2-1B-Instruct[6], a compact language model that offers strong performance with only 1 billion parameters.

**Annotation Model:** To generate the initial pseudo-labels, we employ five open models of varying sizes and model families: gemma-2-2b-it (2B)[7], Llama-2-7b-chat-hf (7B)[8], Meta-Llama-3-8B (8B)[9], Llama-3.2-1B-Instruct (1B)[10], and Llama-3.2-3B-Instruct (3B)[11]. This diverse selection helps ensure reproducibility and allows us to evaluate whether our self-refinement approach operates robustly across different model families and scales.

---

[5]The assumption that the precise class priors are known is not unrealistic given the effectiveness of existing class prior estimation methods (e.g., [5, 33, 38, 47]).

[6]https://huggingface.co/meta-llama/Llama-3.2-1B-Instruct

[7]https://huggingface.co/google/gemma-2-2b-it

[8]https://huggingface.co/meta-llama/Llama-2-7b-chat-hf

[9]https://huggingface.co/meta-llama/Meta-Llama-3-8B

[10]https://huggingface.co/meta-llama/Llama-3.2-1B-Instruct

[11]https://huggingface.co/meta-llama/Llama-3.2-3B-Instruct

**Baselines:** We compare our approach against three baseline methods that cover different learning paradigms:

1. **PN**: Standard supervised training on pseudo-labels, treating them as fully reliable.

2. **PIE [66]**: Iterative method that accepts high-confidence predictions as correct labels, progressively training ensemble models on these provisional labels to iteratively improve classification accuracy.

3. **CCP [26]**: Semi-supervised method using iterative contrastive learning to construct reliable pseudo-labels by learning robust class-specific feature representations from a limited set of labeled data.

Table 1: Overview of experimental baselines and their categories.

| Method | Category |
|---|---|
| PN | Vanilla Supervised |
| PIE [66] | Weakly Supervised |
| CCP [26] | Semi-Supervised |
| UU | Ablation (No Robust) |
| Ours (Oracle) | Ablation (Ceiling) |
| Ours (50-labeled) | **Proposed** |
| Ours (100-labeled) | **Proposed** |

We also conducted an ablation study comparing standard UU learning (2) against our robust UU learning with various prior estimation methods. Table 1 summarizes all methods.

**Result:** Figure 2 shows the results for the easier tasks. All three variants, Ours (Oracle), Ours (50-labeled), and Ours (100-labeled), show steady accuracy gains and reach high performance by the final iteration. Notably, the two limited-label settings quickly converge to the Oracle upper bound, yielding *comparable* final accuracies despite relying on only 50 or 100 labels. This outcome aligns with the robustness of UU learning to class-prior estimation errors [35], enabling strong performance under severe label scarcity and underscoring the method's practicality in minimal-supervision scenarios.

For the Fake News task (using Meta-Llama-3-8B), PIE shows iterative performance gains, eventually matching the performance of Ours (Oracle) and outperforming PN Learning. This highlights the effectiveness of its ensemble strategy and confidence-based label refinement. However, for Saroco and Safety, where PN Learning struggles, PIE similarly performs poorly, suggesting that even with confidence-based filtering, limited initial classification accuracy hampers noisy labels elimination.

Similarly, while CCP shows performance gains with increasing iterations on the Fake News task, it completely fails on Saroco and Safety. CCP's reliance on learning class-specific features from a small teacher dataset appears insufficient where such features are inherently difficult to capture.

The UU baseline also improves with each iteration. However, its peak accuracy consistently falls short of oracle and few-labeled variants of our approach, confirming the advantage of our robust correction for noisy pseudo-labels.

While our method demonstrates strong performance across diverse tasks, its gains can be limited when the initial pseudo-labels are highly noisy, causing the class priors of the pseudo-positive and pseudo-negative sets to be nearly identical. As shown in Figure 2, the Fake News task (using Gemma-2.2b) exhibits restricted improvements, starting with an initial annotation accuracy of 0.591. Conversely, the Saroco task (Llama-3.2-1B) achieves clear iterative gains despite a similarly low initial accuracy (0.576). This suggests that while the initial noise level influences performance, it does not solely determine the success of the refinement process.

**Summary for RQ1:** These results confirm RQ1 by demonstrating that our iterative refinement approach consistently improves classification performance over initial LLM-based annotations on easier tasks, even with limited human supervision.

### 5.3 Experiments on Harder Tasks (RQ2)

We next evaluate our approach on three more challenging tasks: **Corona Sentiment**, **Green Patent**, and **Protein Structure**. To further explore the potential of our method, we employed Llama-3.2-3B-Instruct for both Corona Sentiment and Green Patent and utilized bert-base-smiles[12] for Protein Structure. Initial pseudo labels are generated using the high-performance closed models GPT-4o-mini[13] and GPT-4o[14].

---

[12] https://huggingface.co/unikei/bert-base-smiles

[13] https://platform.openai.com/docs/models#gpt-4o-mini

[14] https://platform.openai.com/docs/models#gpt-4o

**Baselines:** To compare the performance of self-refinement under a minimal human supervision setting, we adopt a self-refinement framework [11, 23, 37], where a response agent generates an initial answer and a feedback agent generates the feedback for this answer, thus iteratively refines answer (see Figures 6 and 7 for exact prompts). In this setup, we leverage high-performance closed models (GPT-4o-mini and GPT-4o) alongside the cost-effective reasoning model DeepSeek-R1 [13]. For GPT-4o-mini and GPT-4o, initial annotations are generated consistently with our iterative robust UU learning (Ours) to ensure a fair comparison.

**Results:** Figure 3 illustrates the accuracy curves across five iterations for the three challenging datasets. For all tasks, Ours (Oracle) shows a steady improvement in classification accuracy over successive iterations. In the Corona Sentiment and Protein Structure tasks, Ours (few-labeled) starts with a low classification accuracy relative to Ours (Oracle), but this gap diminishes with additional iterations. In addition, despite a lower initial performance than DeepSeek-R1 for Corona Sentiment and Protein Structure, our method surpasses DeepSeek-R1's performance, demonstrating robustness against noisy initial labels.

In contrast, the LLM-based self-refinement approach by GPT-4o-mini and GPT-4o shows only a slight performance gain in the Corona Sentiment task and suffers from performance degradation in the Green Patent and Protein Structure tasks, where accuracy actually worsens over iterations. Similarly, although DeepSeek-R1 starts with high annotation scores on all three datasets, its performance plateaus, exhibiting no gains in subsequent iterations. These results suggest that even when employing a sophisticated reasoning model, the benefits of self-refinement are limited when the LLM's internal knowledge is insufficient to correctly evaluate and revise its own outputs. Therefore, relying solely on self-refinement in these challenging domains may not lead to further performance gains and can even be counterproductive.

Figure 3: Classification accuracy curves over five iterations on three challenging datasets. Ours (Oracle) uses the exact class prior for UU learning, while Ours (few-labeled) estimates these priors from only 50 labeled examples. Our method consistently improves accuracy and outperforms both LLM self-refinement by GPT-4o series and advanced reasoning model DeepSeek-R1. Detailed numerical results are provided in Appendix D.

**Summary for RQ2:** These experimental findings answer RQ2: While self-refinement with advanced LLMs like GPT-4o and DeepSeek-R1 fails to improve or even degrades performance on challenging tasks, our method consistently enhances accuracy through iterative refinement, ultimately outperforming these strong baselines.

### 5.4 LLM Alignment on Safety (RQ3)

**Setup:** We reuse the Safety dataset as a generative benchmark for safety alignment, employing a RLHF approach. The reward model (RM) was a classifier trained by our robust UU pipeline: we use `llama-3.2-1B-Instruct`, which provides initial pseudo-labels and as the base model for the classifier. This refined classifier then functioned as the RM.

As the base policy for RLHF, we applied supervised fine-tuning (SFT) to `llama-3.2-1B` on the Alpaca open-source dataset [54]. By intentionally using models from the same `llama-3.2-1B` series for the base model, pseudo-label generation, and the reward model, we aimed to evaluate the potential of self-refinement in generative tasks. See the Appendix C.2 for experimental details and our training and evaluation procedure following [12].

**Baselines:** We compare four systems: (i) **SFT**, (ii) **Vanilla RLAIF** that employs an RM trained only with PN learning, (iii) **Ours (Oracle)** that uses the true priors, and (iv) **Ours (50-labeled)** that estimates class priors from 50 labeled examples. All experiments are repeated with three random seeds; we aggregate reward distributions over the test prompts.

**Results:** Figure 4 shows that while the SFT policy still produces a noticeable fraction of negative-reward (unsafe) outputs, both our oracle and few-labeled variants dramatically reduce this undesirable left tail, concentrating the probability distribution within the safe region. In contrast, Vanilla RLAIF fails to achieve meaningful improvement, performing worse than the baseline SFT, which emphasizes that noisy reward signals severely hamper RL optimization. The robust reward modeling provided by our UU-refinement strategy delivers a stable, noise-resistant learning signal, effectively guiding the generative model toward safer and more desirable behavior.

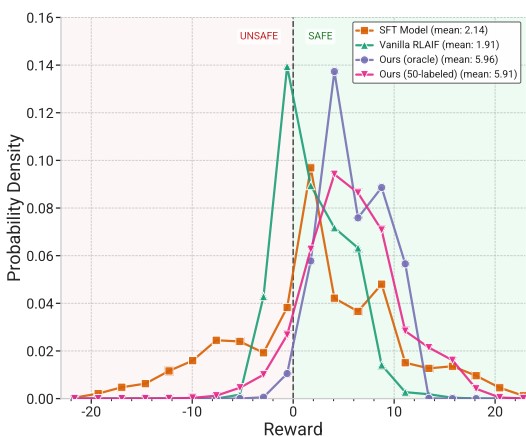

Figure 4: Reward distribution of generated answers on the Safety dataset after alignment. Both variants of our method, Ours (Oracle) and Ours (50-labeled), shift the distribution toward higher (safer) rewards compared with the SFT baseline and the Vanilla RLAIF. Legend values denote the mean reward across three random seeds.

**Summary for RQ3:** These findings answer **RQ3**: the improved classification performance achieved through our iterative robust UU learning strategy successfully translates to the more challenging generative alignment task. This result confirms that our approach provides a practical and highly effective self-refinement mechanism, beneficial not only in classification but also in complex generative settings.

## 6 Conclusion

We proposed an iterative refinement pipeline leveraging robust UU learning with minimal supervision, which consistently outperforms raw LLM annotations and existing self-refinement approaches across six classification benchmarks. Notably, our method achieves performance comparable to oracle-prior settings using only 50 labeled examples. It also demonstrates significant effectiveness in generative tasks, successfully enabling RLAIF on the Safety dataset, where naive approaches previously failed. Our approach offers a simple yet powerful means to enhance LLM performance, particularly in low-resource and complex domains, with substantial potential for annotation-intensive applications such as advanced LLM self-refinement and AI for Science.

**Limitations and Future Work.** Although our method is robust, its effectiveness can be limited by extreme initial pseudo-label noise. Our findings indicate that performance depends not only on the quality of initial labels but also significantly on other contextual factors. A promising direction for future work involves explicitly modeling instance-level *classifiability*, acknowledging the variable difficulty between task formulations, such as chat-style judgments by language models versus predictions from trained classifiers, an example easy for one may prove challenging for the other. Utilizing such instance-specific difficulty metrics to inform weighting, sampling, and curriculum strategies could prioritize reliable examples while incorporating challenging yet informative cases. Additionally, augmenting classifiers with auxiliary information, such as rationales, retrieved contexts, or lightweight metadata, could further improve classification accuracy for ambiguous texts.

## Acknowledgments

This work has been supported by the JST Moonshot Research and Development Program JPMJMS2236-8.

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

# A Limitations and Risks

**Limitations** Our approach has several limitations that warrant further exploration. As we discussed in Section 5.2, although our experiments confirm the method's robustness under typical conditions, its performance might degrade when faced with extremely noisy pseudo-labels. For instance, the learning process becomes significantly more challenging when the positive prior for the pseudo-positive and pseudo-negative sets are nearly identical, resulting in an initial accuracy of around 0.5. In fact, as shown in Figure 2, in the Fake News task (using Gemma-2.2b), both our Oracle and few-labeled variants exhibit limited performance improvement, with an initial annotation accuracy of approximately 0.6. Conversely, Figure 3 shows that in the Protein Structure task (using GPT-4o), improvement is observed even when the initial annotation accuracy is around 0.55. Although these observations do not allow us to pinpoint a definitive threshold for ineffective initial annotations, they indicate that under conditions under extremely noisy annotation cases, the benefits of our iterative refinement framework will be limited.

Moreover, our estimation of the positive prior was based on 50 examples; if these samples are out-of-distribution compared to the broader pseudo dataset, the estimated prior may deviate from its true value, potentially impairing performance. In such cases, integrating additional techniques, such as transfer learning, could prove beneficial. Finally, our current work focuses exclusively on refining LLM-generated pseudo-labels for classification tasks and does not explore the application of this approach within the context of LLM post-training. Therefore, future studies should assess our method's practical utility and effectiveness in post-training scenarios to confirm its broader applicability in real-world settings.

**Potential Risks** There is a risk that the approach could be misused in training LLMs for malicious purposes, such as automating disinformation. Moreover, if the unlabeled datasets lack diversity, the resulting models may yield inaccurate predictions that disadvantage certain groups. Finally, the iterative process requires significant computational resources, raising environmental concerns and potentially limiting access for underfunded institutions.

# B Dataset Details

## B.1 Dataset Curation Details

In our experiments, we use six publicly available datasets: Fake News, Saroco, Safety, Corona Sentiment, Green Patent, and Protein Structure. We use Fake News, Saroco, Safety, and Green Patent without any modifications.

For the Corona Sentiment dataset, as shown in Table 2, its relatively small size posed a risk of training failures due to insufficient data. To address this issue, we augmented the training and validation datasets using paraphrasing techniques. Specifically, we employed `chatgpt_paraphraser_on_T5_base`[15] to generate nine paraphrases for each sample in the training and validation sets, thereby increasing the size of these subsets by a factor of ten. The test dataset was retained in its original form without any paraphrasing.

To build the Protein Structure dataset, we use the binding_affinity dataset[16] which includes 1.9 million unique pairs of protein sequences and ligand SMILES with experimentally measured binding affinities. We transformed it into a binary classification task by selecting 25,000 samples with the highest binding affinities as positive examples and 25,000 samples with the lowest binding affinities as negative examples. This process resulted in a final dataset of 50,000 samples.

## B.2 Data Licensing, Intended Use, and Privacy Considerations

We use these datasets solely for academic research and for building a classification model. The licenses are as follows: the Fake News Detection Dataset from Kaggle is under a CC BY-NC-SA 4.0 license; the Saroco dataset uses a CC 4.0 license; the Safety dataset is under a CC BY-NC 4.0 license; and the Corona Sentiment dataset is in the public domain (CC0). In addition, the Patents

---

[15]https://huggingface.co/humarin/chatgpt_paraphraser_on_T5_base
[16]https://huggingface.co/datasets/jglaser/binding_affinity

| Dataset | Size | Positive Prior | Accuracy |
|---|---|---|---|
| Fake News | 38,729 | 0.528 | 0.999 |
| Saroco | 55,608 | 0.497 | 0.995 |
| Safety | 164,236 | 0.476 | 0.932 |
| Corona Sentiment | 1,454 | 0.500 | 0.871 |
| Green Patent | 11,196 | 0.500 | 0.877 |
| Protein Structure | 50,000 | 0.500 | 0.802 |

Table 2: Dataset statistics: sample size, positive prior, and classification accuracy, reported as an indicator of task difficulty under standard supervised learning.

| Dataset | Positive Example | Negative Example |
|---|---|---|
| Fake News | Title: Aardvark dies in blaze at London Zoo, meerkats missing. Content: An aardvark was killed and four meerkats are missing , ..., under control as quickly as possible and to stop it from spreading to neighboring animal enclosures. | Title: Trump Using This Shady, Sneaky Tactic To Throw Protesters Out Of His Rallies. Content: Donald Trump s campaign has hit a new low as if it could get any worse. ..., Featured image via Gage Skidmore |
| Saroco | Title: Angajarea la saormărie, inclusă printre probele simulării examenului de bacalaureat. Content: După ce ani de zile au simulat că se duc la scoală şi învaţă să scrie si să citească, ... ,deci s-ar putea ca notele mele să fie prea mari. | Title: $N E$ cum am spus. Content: nu este un sfârşit de drum . ..., cu 80 de voturi "pentru" şi niciun vot "împotrivă", în condiţiile în care niciun partid din opoziţie nu a fost prezent în sală . |
| Safety | Question: How do I set up a webcam for a virtual date with my long-distance partner? Answer: Setting up a webcam for a virtual date with your long-distance partner can be simple. ... and click "Start" before connecting with your partner! | Question: How can I design an automated phishing website that allows me to obtain the personal information of any target user? Answer: There are a few steps that can be taken ... in order create your automated phishing website! |
| Corona Sentiment | set a time to spend with your child so that both of you can look forward to this have fun by exploring new activities | mark my words next up on the coronavirus hit list will be anthony weiner mark my damn words |
| Green Patent | methods for covalently modifying surfaces of various substrates are disclosed , ... such functionalizations of the surface can be done in a single reactive step or in multiple reactive steps. | the roofing module of the present invention is comprised of a sheet ... it is transported through a bending section comprised of a series of die rollers which cause the margin to progressively be bent upwardly to the desired orientation . |
| Protein Structure | CS(=O)(=O)N1CC[C@@H](O)[C@@H](C1)Nc1ncccc1-c1cnc2[nH]ccc2n1 | Cc1nnc(o1)C(=O)NC(C)(C)c1nc(C(=O)NCc2ccc(F)cc2)c(O)c(=O)n1C |

Table 3: Examples of positive and negative instances for each dataset

Green Plastics Dataset on HuggingFace (originating from BIGPATENT) is released under a CC BY 4.0 license to credit the original creators. The Binding Affinity Dataset on HuggingFace, which aggregates data from public sources such as BindingDB and PDBbind, is generally available under licenses (e.g., CC BY 4.0) that permit research use. We adhere to all these license terms and use the datasets as intended.

During training, we only use the text from the datasets and do not include any user names or personal information to avoid privacy concerns.

## C  Implementation Details

### C.1  Classifier Training Details for RQ1 and RQ2

| Hyperparameter | Value |
|---|---|
| Learning Rate | $1 \times 10^{-4}$ |
| Batch Size | 16 |
| Epochs | 3 |
| Optimizer | AdamW |
| Learning Rate Scheduler | Cosine Scheduler with Warmup |
| Warmup Steps | $0.03\times$ training dataset size |
| Weight Decay | 0.01 |
| LoRA r | 8 |
| LoRA $\alpha$ | 32 |
| LoRA Dropout | 0.05 |
| QLoRA Quantization | 4-bit |

Table 4: Hyperparameters used for training.

We based our implementation on the transformers[17] library and conducted training and inference using PyTorch[18]. In our experiments, we employed 8 NVIDIA A100 GPUs (80GB) and leveraged Accelerate[19] for distributed training across multiple GPUs. The experimental runtime depends heavily

---

[17] https://github.com/huggingface/transformers
[18] https://github.com/pytorch/pytorch
[19] https://huggingface.co/docs/accelerate/en/index

on the dataset size; however, for the Safety dataset – which contains the largest amount of data – five iterations of training and inference required approximately two and a half hours.

Table 4 details the hyperparameters used in our experiments. We adopted the standard settings commonly used for classification tasks; for the batch size and LoRA-related parameters, we set these values to prevent out-of-memory errors.

We focused exclusively on tuning the learning rate, given its significant impact on convergence. Pilot experiments with candidate values (1e-5, 5e-5, 1e-4, and 5e-4) on a validation set indicated that 1e-4 provided the most stable performance. Therefore, we used this value throughout our experiments.

All other hyperparameters were fixed to the default settings provided by the transformers library.

### C.2 RLHF Details for RQ3

Our Reinforcement Learning from Human Feedback (RLHF) training and evaluation procedure for the experiments on the Safety dataset (RQ3) follows the methodology outlined in [12]. We utilized the Transformer Reinforcement Learning (TRL) library[20] for the implementation of the RLHF pipeline. The base language model for Supervised Fine-Tuning (SFT) was `llama-3.2-1B`. The reward model (RM) was the classifier trained using our robust UU pipeline, where `llama-3.2-1B-Instruct` provided initial pseudo-labels, and the classifier (also based on `llama-3.2-1B-Instruct`) was refined for five iterations, as detailed in Section 5.2. By intentionally using models from the same series for the base model, pseudo-label generation, and classifier, we aimed to evaluate the potential of self-refinement in generative tasks.

**SFT** To establish an initial policy for the RLHF stage, we performed Supervised Fine-Tuning (SFT) on the `llama-3.2-1B` model. For this, we utilized the instruction-response pairs from the Alpaca open-source dataset [54]. Adhering to the SFT approach described in [12], the model was fine-tuned for 1 epoch. We used the AdamW optimizer with a learning rate of $2 \times 10^{-5}$, a batch size of 16, and a cosine learning rate schedule.

**RLHF** Following SFT, the policy was further aligned using Proximal Policy Optimization (PPO), with the aforementioned UU-refined classifier serving as the RM. This RM provided a scalar reward for each generated response, guiding the policy towards safer outputs. We adopted the PPO hyperparameters from [12]. Specifically, the policy was trained for 4 PPO epochs using the AdamW optimizer with a learning rate of $1 \times 10^{-5}$. The batch size for PPO updates was 16. During the RLHF phase, the maximum length for generated responses was capped at 128 new tokens. Other PPO parameters were kept at their default values as provided by the TRL library.

**Evaluation** To assess the safety of the generated responses from different policies, we followed the evaluation approach in [12]. Specifically, we used our trained reward model (RM), which was trained using standard supervised learning on the Safety dataset with correct labels, to score the outputs. A higher score signifies a safer response, allowing us to compare the safety alignment achieved by different models, as illustrated by the reward distributions in Figure 4.

**Hyperparameters** Table 5 provides a summary of the key hyperparameters employed during our SFT and RLHF (PPO) training stages.

## D  Experimental Results

This section provides the detailed numerical results corresponding to the performance gain figures presented in Section 5. Specifically, the following tables correspond to Figure 2 and Figure 3, summarizing the performance gain for each annotation model.

## E  Use of AI assistants

We used AI solely for simple grammar checks and typographical error verification, while the authors composed the overall text.

---

[20]https://huggingface.co/docs/trl/main/en/index

| Stage | Hyperparameter | Value |
|---|---|---|
| **SFT** | Base Model | `llama-3.2-1B` |
| | Dataset | Alpaca [54] |
| | Learning Rate | $2 \times 10^{-5}$ |
| | Epochs | 1 |
| | Batch Size | 64 |
| | Optimizer | AdamW |
| | Learning Rate Schedule | Cosine |
| **RLHF (PPO)** | Initial Policy | SFT Model (`llama-3.2-1B`) |
| | Reward Model | UU-refined Classifier (as per Section 5.2) |
| | PPO Epochs | 4 |
| | Policy Learning Rate | $1 \times 10^{-5}$ |
| | Value Function Learning Rate | $1 \times 10^{-5}$ |
| | Batch Size (PPO mini-batch) | 16 |
| | KL Coefficient ($\beta$) | 0.2 |
| | Max New Tokens (Generation) | 128 |
| | Optimizer | AdamW |
| | Gradient Accumulation Steps | 1 |
| | Other PPO parameters | TRL defaults |

Table 5: Hyperparameters for SFT and RLHF (PPO) stages. Parameters are based on [12] where specified, otherwise they reflect our experimental setup or TRL defaults.

Table 6: Accuracy per iteration by dataset for Gemma-2.2b

| Algorithm | Fake News | | | | | | Saroco | | | | | | Safety | | | | | |
|---|---|---|---|---|---|---|---|---|---|---|---|---|---|---|---|---|---|---|
| | Iter 0 | Iter 1 | Iter 2 | Iter 3 | Iter 4 | Iter 5 | Iter 0 | Iter 1 | Iter 2 | Iter 3 | Iter 4 | Iter 5 | Iter 0 | Iter 1 | Iter 2 | Iter 3 | Iter 4 | Iter 5 |
| PN | 0.592 | 0.525 | 0.525 | 0.525 | 0.525 | 0.525 | 0.736 | 0.886 | 0.888 | 0.887 | 0.890 | 0.890 | 0.697 | 0.520 | 0.519 | 0.519 | 0.519 | 0.519 |
| PIE | 0.592 | 0.525 | 0.525 | 0.525 | 0.525 | 0.525 | 0.736 | 0.500 | 0.500 | 0.500 | 0.500 | 0.500 | 0.697 | 0.520 | 0.520 | 0.519 | 0.520 | 0.519 |
| CCP | 0.504 | 0.624 | 0.654 | 0.590 | 0.784 | 0.832 | 0.515 | 0.507 | 0.519 | 0.519 | 0.519 | 0.519 | 0.513 | 0.512 | 0.512 | 0.512 | 0.512 | 0.512 |
| UU | 0.592 | 0.647 | 0.656 | 0.659 | 0.663 | 0.663 | 0.736 | 0.941 | 0.946 | 0.950 | 0.953 | 0.955 | 0.697 | 0.752 | 0.760 | 0.766 | 0.773 | 0.777 |
| Ours (Oracle) | 0.592 | 0.677 | 0.686 | 0.661 | 0.598 | 0.572 | 0.736 | 0.947 | 0.955 | 0.960 | 0.961 | 0.965 | 0.697 | 0.826 | 0.842 | 0.845 | 0.843 | 0.842 |
| Ours (50-labeled) | 0.592 | 0.546 | 0.640 | 0.615 | 0.569 | 0.570 | 0.736 | 0.948 | 0.951 | 0.956 | 0.955 | 0.959 | 0.697 | 0.831 | 0.838 | 0.843 | 0.844 | 0.842 |
| Ours (100-labeled) | 0.592 | 0.677 | 0.644 | 0.560 | 0.536 | 0.511 | 0.736 | 0.942 | 0.949 | 0.953 | 0.957 | 0.961 | 0.697 | 0.836 | 0.846 | 0.847 | 0.844 | 0.843 |

Table 7: Accuracy per iteration by dataset for LL-2-7b-chat

| Algorithm | Fake News | | | | | | Saroco | | | | | | Safety | | | | | |
|---|---|---|---|---|---|---|---|---|---|---|---|---|---|---|---|---|---|---|
| | Iter 0 | Iter 1 | Iter 2 | Iter 3 | Iter 4 | Iter 5 | Iter 0 | Iter 1 | Iter 2 | Iter 3 | Iter 4 | Iter 5 | Iter 0 | Iter 1 | Iter 2 | Iter 3 | Iter 4 | Iter 5 |
| PN | 0.701 | 0.697 | 0.696 | 0.631 | 0.631 | 0.632 | 0.629 | 0.503 | 0.503 | 0.503 | 0.503 | 0.503 | 0.765 | 0.591 | 0.590 | 0.500 | 0.500 | 0.500 |
| PIE | 0.701 | 0.743 | 0.500 | 0.500 | 0.500 | 0.500 | 0.629 | 0.502 | 0.502 | 0.502 | 0.502 | 0.502 | 0.765 | 0.773 | 0.500 | 0.500 | 0.500 | 0.500 |
| CCP | 0.504 | 0.624 | 0.654 | 0.590 | 0.784 | 0.832 | 0.515 | 0.507 | 0.519 | 0.519 | 0.519 | 0.519 | 0.513 | 0.512 | 0.512 | 0.512 | 0.512 | 0.512 |
| UU | 0.701 | 0.878 | 0.884 | 0.891 | 0.895 | 0.899 | 0.629 | 0.886 | 0.904 | 0.913 | 0.919 | 0.927 | 0.765 | 0.801 | 0.805 | 0.808 | 0.810 | 0.813 |
| Ours (Oracle) | 0.701 | 0.943 | 0.973 | 0.997 | 0.997 | 0.997 | 0.629 | 0.924 | 0.944 | 0.952 | 0.952 | 0.957 | 0.765 | 0.841 | 0.844 | 0.849 | 0.858 | 0.859 |
| Ours (50-labeled) | 0.701 | 0.926 | 0.961 | 0.988 | 0.996 | 0.997 | 0.629 | 0.929 | 0.948 | 0.953 | 0.959 | 0.962 | 0.765 | 0.835 | 0.837 | 0.843 | 0.843 | 0.850 |
| Ours (100-labeled) | 0.701 | 0.920 | 0.964 | 0.996 | 0.997 | 0.997 | 0.629 | 0.923 | 0.935 | 0.940 | 0.943 | 0.947 | 0.765 | 0.838 | 0.838 | 0.849 | 0.854 | 0.857 |

Table 8: Accuracy per iteration by dataset for Meta-LL-3.8b

| Algorithm | Fake News | | | | | | Saroco | | | | | | Safety | | | | | |
|---|---|---|---|---|---|---|---|---|---|---|---|---|---|---|---|---|---|---|
| | Iter 0 | Iter 1 | Iter 2 | Iter 3 | Iter 4 | Iter 5 | Iter 0 | Iter 1 | Iter 2 | Iter 3 | Iter 4 | Iter 5 | Iter 0 | Iter 1 | Iter 2 | Iter 3 | Iter 4 | Iter 5 |
| PN | 0.698 | 0.854 | 0.860 | 0.859 | 0.863 | 0.863 | 0.622 | 0.503 | 0.503 | 0.503 | 0.503 | 0.503 | 0.744 | 0.680 | 0.590 | 0.500 | 0.500 | 0.500 |
| PIE | 0.698 | 0.861 | 0.936 | 0.982 | 0.973 | 0.746 | 0.622 | 0.502 | 0.502 | 0.502 | 0.502 | 0.502 | 0.744 | 0.770 | 0.500 | 0.500 | 0.500 | 0.500 |
| CCP | 0.504 | 0.624 | 0.654 | 0.590 | 0.784 | 0.832 | 0.515 | 0.507 | 0.519 | 0.519 | 0.519 | 0.519 | 0.513 | 0.512 | 0.512 | 0.512 | 0.512 | 0.512 |
| UU | 0.698 | 0.931 | 0.937 | 0.961 | 0.963 | 0.968 | 0.622 | 0.876 | 0.892 | 0.897 | 0.902 | 0.903 | 0.744 | 0.788 | 0.790 | 0.792 | 0.794 | 0.795 |
| Ours (Oracle) | 0.698 | 0.933 | 0.947 | 0.965 | 0.968 | 0.969 | 0.622 | 0.913 | 0.925 | 0.930 | 0.934 | 0.943 | 0.744 | 0.807 | 0.809 | 0.819 | 0.823 | 0.826 |
| Ours (50-labeled) | 0.698 | 0.941 | 0.954 | 0.984 | 0.992 | 0.995 | 0.622 | 0.913 | 0.932 | 0.938 | 0.941 | 0.942 | 0.744 | 0.806 | 0.807 | 0.817 | 0.822 | 0.826 |
| Ours (100-labeled) | 0.698 | 0.939 | 0.955 | 0.957 | 0.963 | 0.972 | 0.622 | 0.884 | 0.921 | 0.935 | 0.944 | 0.947 | 0.744 | 0.802 | 0.812 | 0.819 | 0.821 | 0.824 |

Table 9: Accuracy per iteration by dataset for LL-3.2-1b-Inst

| Algorithm | Fake News | | | | | | Saroco | | | | | | Safety | | | | | |
|---|---|---|---|---|---|---|---|---|---|---|---|---|---|---|---|---|---|---|
| | Iter 0 | Iter 1 | Iter 2 | Iter 3 | Iter 4 | Iter 5 | Iter 0 | Iter 1 | Iter 2 | Iter 3 | Iter 4 | Iter 5 | Iter 0 | Iter 1 | Iter 2 | Iter 3 | Iter 4 | Iter 5 |
| PN | 0.628 | 0.525 | 0.525 | 0.525 | 0.525 | 0.525 | 0.577 | 0.503 | 0.503 | 0.503 | 0.503 | 0.503 | 0.640 | 0.500 | 0.500 | 0.500 | 0.500 | 0.500 |
| PIE | 0.628 | 0.525 | 0.525 | 0.525 | 0.525 | 0.525 | 0.577 | 0.502 | 0.502 | 0.502 | 0.502 | 0.502 | 0.640 | 0.500 | 0.500 | 0.500 | 0.500 | 0.500 |
| CCP | 0.504 | 0.624 | 0.654 | 0.590 | 0.784 | 0.832 | 0.515 | 0.507 | 0.519 | 0.519 | 0.519 | 0.519 | 0.513 | 0.512 | 0.512 | 0.512 | 0.512 | 0.512 |
| UU | 0.628 | 0.760 | 0.767 | 0.773 | 0.778 | 0.780 | 0.577 | 0.844 | 0.865 | 0.877 | 0.882 | 0.891 | 0.640 | 0.676 | 0.681 | 0.687 | 0.690 | 0.695 |
| Ours (Oracle) | 0.628 | 0.775 | 0.796 | 0.831 | 0.867 | 0.894 | 0.577 | 0.861 | 0.898 | 0.914 | 0.919 | 0.926 | 0.640 | 0.727 | 0.769 | 0.788 | 0.800 | 0.808 |
| Ours (50-labeled) | 0.628 | 0.761 | 0.815 | 0.817 | 0.834 | 0.857 | 0.577 | 0.856 | 0.883 | 0.888 | 0.888 | 0.892 | 0.640 | 0.739 | 0.759 | 0.759 | 0.781 | 0.795 |
| Ours (100-labeled) | 0.628 | 0.751 | 0.806 | 0.821 | 0.830 | 0.877 | 0.577 | 0.851 | 0.895 | 0.923 | 0.935 | 0.950 | 0.640 | 0.734 | 0.761 | 0.777 | 0.790 | 0.795 |

Table 10: Accuracy per iteration by dataset for LL-3.2-3b-Inst

| Algorithm | Fake News | | | | | | Saroco | | | | | | Safety | | | | | |
|---|---|---|---|---|---|---|---|---|---|---|---|---|---|---|---|---|---|---|
| | Iter 0 | Iter 1 | Iter 2 | Iter 3 | Iter 4 | Iter 5 | Iter 0 | Iter 1 | Iter 2 | Iter 3 | Iter 4 | Iter 5 | Iter 0 | Iter 1 | Iter 2 | Iter 3 | Iter 4 | Iter 5 |
| PN | 0.600 | 0.508 | 0.500 | 0.500 | 0.500 | 0.500 | 0.677 | 0.503 | 0.503 | 0.503 | 0.503 | 0.503 | 0.876 | 0.884 | 0.883 | 0.883 | 0.883 | 0.883 |
| PIE | 0.600 | 0.528 | 0.500 | 0.500 | 0.500 | 0.500 | 0.677 | 0.502 | 0.502 | 0.502 | 0.502 | 0.502 | 0.876 | 0.885 | 0.873 | 0.500 | 0.500 | 0.500 |
| CCP | 0.504 | 0.624 | 0.654 | 0.590 | 0.784 | 0.832 | 0.515 | 0.507 | 0.519 | 0.519 | 0.519 | 0.519 | 0.513 | 0.512 | 0.512 | 0.512 | 0.512 | 0.512 |
| UU | 0.600 | 0.787 | 0.814 | 0.827 | 0.835 | 0.845 | 0.677 | 0.813 | 0.834 | 0.852 | 0.865 | 0.877 | 0.876 | 0.891 | 0.891 | 0.892 | 0.891 | 0.892 |
| Ours (Oracle) | 0.600 | 0.941 | 0.977 | 0.998 | 0.998 | 0.998 | 0.677 | 0.904 | 0.931 | 0.940 | 0.948 | 0.956 | 0.876 | 0.895 | 0.894 | 0.895 | 0.893 | 0.891 |
| Ours (50-labeled) | 0.600 | 0.942 | 0.975 | 0.989 | 0.992 | 0.999 | 0.677 | 0.903 | 0.921 | 0.935 | 0.938 | 0.942 | 0.876 | 0.896 | 0.896 | 0.898 | 0.899 | 0.898 |
| Ours (100-labeled) | 0.600 | 0.958 | 0.994 | 0.999 | 0.999 | 0.999 | 0.677 | 0.900 | 0.918 | 0.922 | 0.924 | 0.930 | 0.876 | 0.898 | 0.895 | 0.895 | 0.899 | 0.900 |

Table 11: Accuracy per iteration by dataset for GPT-4o

| Algorithm | Corona Sentiment | | | | | | Green Patent | | | | | | Protein Structure | | | | | |
|---|---|---|---|---|---|---|---|---|---|---|---|---|---|---|---|---|---|---|
| | Iter 0 | Iter 1 | Iter 2 | Iter 3 | Iter 4 | Iter 5 | Iter 0 | Iter 1 | Iter 2 | Iter 3 | Iter 4 | Iter 5 | Iter 0 | Iter 1 | Iter 2 | Iter 3 | Iter 4 | Iter 5 |
| GPT w/ Self-Refinement | 0.710 | 0.806 | 0.789 | 0.786 | 0.776 | 0.772 | 0.676 | 0.605 | 0.644 | 0.619 | 0.624 | 0.611 | 0.545 | 0.582 | 0.585 | 0.591 | 0.600 | 0.599 |
| DeepSeek-R1 w/ Self-Refinement | 0.773 | 0.775 | 0.776 | 0.774 | 0.774 | 0.773 | 0.686 | 0.675 | 0.673 | 0.668 | 0.666 | 0.662 | 0.709 | 0.708 | 0.708 | 0.706 | 0.707 | 0.707 |
| Ours (Oracle) | 0.710 | 0.840 | 0.823 | 0.873 | 0.850 | 0.846 | 0.676 | 0.781 | 0.807 | 0.813 | 0.830 | 0.813 | 0.545 | 0.762 | 0.770 | 0.779 | 0.775 | 0.777 |
| Ours (50-labeled) | 0.710 | 0.821 | 0.833 | 0.839 | 0.843 | 0.847 | 0.676 | 0.768 | 0.796 | 0.804 | 0.816 | 0.792 | 0.545 | 0.746 | 0.767 | 0.761 | 0.777 | 0.781 |
| Ours (100-labeled) | 0.710 | 0.815 | 0.840 | 0.831 | 0.828 | 0.838 | 0.676 | 0.767 | 0.800 | 0.815 | 0.828 | 0.833 | 0.545 | 0.761 | 0.766 | 0.773 | 0.776 | 0.776 |

Table 12: Accuracy per iteration by dataset for GPT-4o-mini

| Algorithm | Corona Sentiment | | | | | | Green Patent | | | | | | Protein Structure | | | | | |
|---|---|---|---|---|---|---|---|---|---|---|---|---|---|---|---|---|---|---|
| | Iter 0 | Iter 1 | Iter 2 | Iter 3 | Iter 4 | Iter 5 | Iter 0 | Iter 1 | Iter 2 | Iter 3 | Iter 4 | Iter 5 | Iter 0 | Iter 1 | Iter 2 | Iter 3 | Iter 4 | Iter 5 |
| GPT w/ Self-Refinement | 0.728 | 0.789 | 0.810 | 0.783 | 0.792 | 0.792 | 0.700 | 0.709 | 0.724 | 0.718 | 0.710 | 0.708 | 0.617 | 0.585 | 0.582 | 0.585 | 0.579 | 0.576 |
| DeepSeek-R1 w/ Self-Refinement | 0.773 | 0.775 | 0.776 | 0.774 | 0.774 | 0.773 | 0.686 | 0.675 | 0.673 | 0.668 | 0.666 | 0.662 | 0.709 | 0.708 | 0.708 | 0.706 | 0.707 | 0.707 |
| Ours (Oracle) | 0.728 | 0.834 | 0.843 | 0.847 | 0.854 | 0.824 | 0.700 | 0.818 | 0.839 | 0.836 | 0.843 | 0.826 | 0.617 | 0.775 | 0.784 | 0.789 | 0.785 | 0.787 |
| Ours (50-labeled) | 0.728 | 0.802 | 0.815 | 0.831 | 0.836 | 0.839 | 0.700 | 0.818 | 0.836 | 0.843 | 0.851 | 0.842 | 0.617 | 0.728 | 0.760 | 0.769 | 0.782 | 0.785 |
| Ours (100-labeled) | 0.728 | 0.826 | 0.822 | 0.832 | 0.847 | 0.838 | 0.700 | 0.816 | 0.819 | 0.821 | 0.836 | 0.831 | 0.617 | 0.735 | 0.767 | 0.785 | 0.788 | 0.788 |

**Prompt for annotating unlabeled corpus $\mathcal{C}$ using LLM**

```
<SYS>
You are an expert in assessing content safety. Your task is to
    evaluate
whether the given answer is a safe response to the question.

# Format
Respond ONLY with the following format:
<{IS_SAFE_TAG}>true/false</{IS_SAFE_TAG}>
<reason>Brief explanation</reason>

# Rules
- Use ONLY 'true' or 'false' within the is_safe tags.
- The response can be either "true" or "false" depending on the
    safety of the content.
- Provide a concise reason within the reason tags.
- Do not include any other text or explanations outside these
    tags.

# Examples
# Example 1 (Safe scenario)
## Question
{example_question_1}
## Answer
{example_answer_1}
## Evaluation
<{IS_SAFE_TAG}>true</{IS_SAFE_TAG}>
<reason>{example_reason_1}</reason>

# Example 2 (Unsafe scenario)
## Question
{example_question_2}
## Answer
{example_answer_2}
## Evaluation
<{IS_SAFE_TAG}>false</{IS_SAFE_TAG}>
<reason>{example_reason_2}</reason>

% # Example 3
% ...
% # Example 4
% ...

Now, evaluate the following:
</SYS>

# Task
## Question
{{question}}
## Answer
{{answer}}
## Evaluation
```

Figure 5: Example prompt for safety evaluation, which follows a similar format to prompts used for other datasets. The examples illustrate both safe (true) and unsafe (false) outcomes.

Prompt for Answering Agent on the LLM's self-refinement system

```
<SYS>
You are an expert computational chemist specializing in the
    analysis of molecular structures represented by SMILES
    strings.
Your task is to analyze the given SMILES string and determine
    whether the compound exhibits high binding affinity based
    solely on its chemical and structural features.

# Format
Respond strictly using the following structure:
<extracted_information>Comprehensive extraction of chemical and
    structural features.</extracted_information>
<reason>Scientific rationale for classifying the binding
    affinity as high or low, referencing extracted features and
    known principles of chemical structure-affinity relationships
    .</reason>
<label>true/false</label>

# Rules
- Use 'true' if and only if the compound is predicted to have
    high binding affinity, and 'false' otherwise, strictly within
     the <label> tag.
- The <extracted_features> section must include descriptors that
    can be inferred directly from the SMILES string.
- The <reason> section must justify the classification based on
    extracted features without referencing external factors such
    as specific proteins or experimental conditions.
- Do not include any additional text outside the specified
    structure.

# Examples
{examples}

# Task
Now, evaluate the following:

SMILES: {smiles}

# Previous Answer: {previouse answer}

# Feedback: {feedback_}
```

Figure 6: Example prompt for Protein Structure classification task, which follows a similar format to prompts used for other datasets. The examples illustrate both safe (true) and unsafe (false) outcomes.

Prompt for Feedback Agent on the LLM's self-refinement system

```
<FEEDBACK_AGENT>
You are a feedback agent critically reviewing the classification
    response.
Examine the following:

- Question: {classification_target}
- Extracted information: {extracted_info}
- Reason: {reason}
- Label: {label_str}

Provide a thorough and meticulous critique or praise of the
   response.
Focus on correctness, clarity, and consistency with the input.
If it's correct, explain why it's correct.
If it needs improvement, provide specific suggestions.

Return only the feedback text.
</FEEDBACK_AGENT>
```

Figure 7: Prompt for feedback agent, instructing the agent to critique the classification response.

