# OpenReview forum: "Self Iterative Label Refinement via Robust Unlabeled Learning"
_NeurIPS.cc/2025/Conference — NeurIPS 2025 poster_

### Official Review · Reviewer_e1xr · 2025-07-02

**Clarity:** 3
**Significance:** 4
**Originality:** 3
**Rating:** 5
**Confidence:** 4

**Summary:**

This paper proposes an iterative label refinement pipeline incorporating a novel Robust Unlabeled-Unlabeled (UU) learning method to enhance LLM-generated pseudo labels for binary classification tasks with minimal human supervision.

The main contributions are as follows:
1) Introduces a novel integration of iterative pseudo labeling with the proposed Robust UU learning method.
2) Demonstrates the effectiveness of the proposed pipeline across diverse binary classification tasks, showing that it outperforms LLM self-refinement approaches.
3) Shows that the pipeline can also benefit reinforcement learning from human feedback (RLHF) by enabling the training of improved reward models.

**Questions:**

1) Could the method be extended beyond binary classification to improve its applicability to more practical, real-world tasks?
2) I would also expect the inclusion of a fully supervised baseline using true labels as a strong point of reference.

**Ethical Concerns:**

["NO or VERY MINOR ethics concerns only"]

**Limitations:**

Yes

**Quality:**

3

**Strengths And Weaknesses:**

Strengths:
1) The proposed method and its application to RLHF are innovative and demonstrate a novel direction for leveraging unlabeled data.
2) The method shows strong performance across multiple tasks. Its success in the RLHF setting further suggests the potential for learning signals that go beyond human-provided supervision.
3) The paper is well-written, with a clear presentation of the methodology and comprehensive experimental details.

Weaknesses:
1) The method is currently limited to binary classification tasks, which restricts its general applicability.
2) In Figure 2, the proposed method performs poorly on the fake news dataset when using Gemma 2.2B, falling behind both UU and CCP methods in the final stages. It would be helpful if the authors could clarify the cause of this performance drop. Additionally, the figure is difficult to interpret due to overlapping lines.
3) The paper would benefit from a comparison with more recent or state-of-the-art semi-supervised learning methods to better contextualize its contributions.

---

> ### Author Rebuttal · Authors · 2025-07-31
>
> Thanks for the valuable feedback. We appreciate your suggestions and will use them to improve the manuscript. Our responses follow.
>
> ---
>
> ### Weakness 1: Restriction to binary classification
>
> Thank you for this critical point.
>
> Because our primary goal was to study LLM self‑refinement, we first concentrated on a binary‑task to anchor the iterative framework.
>
> > Could the method be extended beyond binary classification to improve its applicability to more practical, real-world tasks?
>
> Regarding your question, the answer is **yes**.
> - **Our framework is not inherently limited to the binary case**.
> - It can be **generalized to multi-class problems via methods like complementary-label learning** [1], whose assumptions closely match those of UU-learning.
> - We will revise our conclusion to discuss the extension to multi-class classification as a promising future direction.
>
> [1] https://dl.acm.org/doi/10.5555/3295222.3295315
>
> ---
>
> ### Weakness 2: Performance on the Fake News Dataset
>
> Thank you for highlighting the one setting in which our method lags behind CCP.
>
> In this experiment Gemma‑2.2B starts from exceptionally noisy pseudo‑labels: Fig. 2 reports just 59.1 % test accuracy, also 58.7 % for train dataset. When label noise is this extreme, Robust‑UU may yield only limited gains.
>
> **Noise alone, however, does not preclude improvement**. For example, starting at 57.3 % accuracy on Protein‑Structure(GPT‑4o) and 57.6 % on Sarcos (Llama‑3.2‑1B), **our pipeline still climbs to ≥ 80 % and > 90 %**, respectively. These results illustrate that **performance can rise from a low initial accuracy**, confirming that the Fake‑News/Gemma scenario is an isolated anomaly rather than a systematic weakness.
>
> Indeed, this run is the sole outlier among 21 dataset–model combinations; in the remaining 20 settings our approach outperforms every baseline, underscoring its broad applicability.
>
> To probe this anomaly, we will analyze embeddings of correctly and incorrectly classified samples to pinpoint issues specific to Gemma/Fake‑News. Furthermore, upon acceptance, **we will release the initial pseudo‑labels to facilitate replication and deeper investigation by future researchers**.
>
> ---
>
> ### Weakness 3: Comparison with State-of-the-Art SSL Methods
>
> Thank you for this valuable suggestion.
>
> - **Could you suggest any specific SSL papers** you consider essential? Your recommendations will help ensure that the camera-ready version cites the most relevant work.
> - In the revision, **we will broaden the related-work section** to include these papers and other modern SSL approaches, thereby better contextualising our contribution.
>
> ---
>
> ### Question: Fully Supervised Baseline
> > I would also expect the inclusion of a fully supervised baseline using true labels as a strong point of reference.
>
> A fully supervised baseline (train-on-true-labels) already appears in Appendix B.2, Table 2. We will surface it in the main text and reference it prominently to make comparisons clearer, alongside the figure revisions.
>
> ---
>
> ### Figure Clarity:
> Figure 2 and 3 will be revised for readability, and we will **add tables in the appendix that present the precise numerical data plotted in each figure**.
>
> ---
>
> Once again, we thank you for your time and constructive comments. We are confident that by incorporating these changes, we can significantly improve the clarity and impact of our paper.

---

> > ### Author Response · Authors · 2025-08-06
> >
> > Thank you for reviewing our rebuttal.
> >
> > At your convenience, could you let us know whether our responses addressed your concerns or if any points would benefit from further clarification? We’re happy to provide follow‑ups within the discussion timeline.
> >
> > Thank you again for your time.

---

> > > ### Comment · Reviewer_e1xr · 2025-08-08
> > >
> > > Thank you for your detailed response.
> > >
> > > Your clarification has addressed my concern regarding Weakness 1.
> > >
> > > Regarding Weakness 2, while I understand it may be an isolated anomaly, including an analysis would help readers better understand the limitations and boundary conditions of your method. Would you consider adding such an analysis in the camera-ready version?
> > >
> > > As for Weakness 3, I believe that including FixMatch as a baseline would strengthen the evaluation and provide valuable context for comparison.

---

> > > > ### Author Response · Authors · 2025-08-09
> > > >
> > > > Thank you for your thorough review and constructive feedback. Below is our detailed response and plan for addressing remaining concerns.
> > > >
> > > > ---
> > > >
> > > > ### Weakness 2
> > > > > Would you consider adding such an analysis in the camera-ready version?
> > > >
> > > > **Yes**. We commit to including the proposed analysis in the camera‑ready version. Due to time constraints during the rebuttal period, we could not complete the task and apologize for the inconvenience.
> > > >
> > > > Specifically, we will:
> > > >
> > > > - **Per‑sample difficulty profiling**: Train a standard supervised classifier and use its logit margins as a proxy for classification difficulty.
> > > >
> > > > - **Decomposition by outcome**: Compare difficulty across TP/FP/FN/TN in the initial labels to see how easily classifiable items are distributed and identify cases where UU learning struggles, e.g., when “easy” items cluster in TP/TN and “hard” items dominate FP/FN.
> > > >
> > > > - **Embedding‑space error structure**: Visualize a standard sentence‑embedding space (e.g., via UMAP/t‑SNE) colored by TP/FP/FN/TN to examine how noisy initialization shapes downstream errors.
> > > >
> > > > **Transparency**: Regardless of outcome, we will release code and the initial pseudo‑labels to support replication and further investigation.
> > > >
> > > > ---
> > > >
> > > > ### Weakness 3 — Related SSL literature (including FixMatch)
> > > >
> > > > Thank you for pointing us to FixMatch.
> > > >
> > > > Because FixMatch was originally proposed in computer vision rather than NLP, it fell outside our initial reference set. We agree that incorporating this line of work will strengthen the manuscript. We will integrate FixMatch and closely related SSL methods in the camera‑ready version.

---

### Official Review · Reviewer_py1Q · 2025-07-03

**Clarity:** 3
**Significance:** 2
**Originality:** 2
**Rating:** 4
**Confidence:** 4

**Summary:**

This paper aims to improve LLMs classification and alignment performance with minimal human supervision, especially when initial pseudo-labels generated by the LLMs are noisy or biased.

Claimed contributions:

* The authors introduce a iterative pipeline that refines LLM-generated pseudo-labels using a robust Unlabeled-Unlabeled learning framework, which progressively denoise and improve label quality with minimal human annotation.

* The method is shown to consistently outperform both direct LLM classification and SOTA self-refinement methods (including advanced models like GPT-4o and DeepSeek-R1) on a range of tasks.

* The refined classifier produced enables effective post-training safety alignment of LLMs in reinforcement learning from AI feedback frameworks, and the approach demonstrates successful self-refinement in generative tasks beyond standard classification.

**Questions:**

see the weaknesses;

**Ethical Concerns:**

["NO or VERY MINOR ethics concerns only"]

**Final Justification:**

The review has clearly addressed my concerns, and I appreciate their valuable contribution to this field. As a result, I change my score to 4.

**Limitations:**

yes

**Paper Formatting Concerns:**

no formatting concerns

**Quality:**

2

**Strengths And Weaknesses:**

Strengths:

* The paper tackles the increasingly relevant issue of reducing human supervision in training and aligning large language models.

* This work integrates robust Unlabeled-Unlabeled learning into the iterative self-refinement pipeline, moving beyond simple confidence filtering or ensemble methods.

* This technique is shown to extend beyond binary classification, enabling more effective generative safety alignment (e.g., for RLHF tasks).


Weaknesses:

* Limited novelty: the core idea of robust UU learning is not original to this work and has been established in prior research.

* The paper’s empirical validation and method are restricted to binary classification tasks only.

* The effectiveness depends on the initial pseudo-labels; if the positive ratios between splits are similar (e.g., due to lack of internal knowledge as authors note), the UU learning framework will not help.

* Although the authors claim broader applicability, the empirical experiments are only focused on binary classification. The effectiveness on more general LLM tasks is not fully explored, and it's unclear how well this approach would transfer to other realistic LLM applications.

---

> ### Author Rebuttal · Authors · 2025-07-31
>
> We appreciate your careful assessment and constructive feedback. Below we respond to each of your concerns and clarify the novelty, scope, and robustness of our work.
>
> ---
> ### Weakness 1: Limited novelty
> As other reviewers (e.g., `Lj7L`, `e1xr`) have remarked, our approach is `innovative` and `novel in this context`, and **[NeurIPS reviewing guidelines](https://neurips.cc/Conferences/2025/ReviewerGuidelines) explicitly acknowledge** that a `novel combination of existing techniques` is a valuable contribution.
>
> Our novel contributions are:
> - **First application to LLM self‑refinement**. We are the first to apply theoretically grounded UU learning to the critical task of LLM self‑refinement, **opening a new avenue for performance gains where existin methods that rely on an LLM’s internal knowledge have struggled**. Figures 2 and 3 demonstrate consistent per‑iteration improvements.
> - **Bridging theory and practice**. By embedding principled weak‑supervision theory in a modern alignment pipeline, we provide an effective tool that boosts performance benchmarks (Figs. 2–3) and directly benefits RLHF (Fig. 4).
> ---
> ### Weakness 2: Restriction to binary classification
> You correctly observe that our empirical validation focuses on binary classification.
>
> Because our goal was to study LLM self‑refinement, we first concentrated on a binary‑task to anchor the iterative framework.
>
> However, **our framework is not inherently limited to the binary case.** It can be **generalized to multi‑class problems via a method like complementary‑label learning** [1], whose assumptions closely match those of UU‑learning. We will update our manuscript to reflect this promising future direction.
>
> [1] https://globals.ieice.org/en_transactions/information/10.1587/transinf.2021EDP7035/_p
>
> ---
>
> ### Weakness 3: Dependence on initial pseudo‑labels
> **Robustness to noisy pseudo‑labels is central to our design**, and both theoretically and empirically confirm it.
>
> - **Theory**. UU risk remains unbiased as long as the pseudo‑positive split’s true‑positive rate exceeds that of the pseudo‑negative split, even when the initial accuracy is near 0.5
> - **Practice**. Starting from extremely noisy labels (Protein‑Structure 57.3 %, Sarcos 57.6 %), **our method reaches ≥ 0.80 and > 0.90 within five iterations**, while GPT‑4o, DeepSeek‑R1, PIE, and CCP plateau or decline (Figs. 2–3).
> - **Low‑knowledge regimes**. Because our pipeline relies on data-centric feature extraction rather than internal model knowledge, it keeps improving precisely even when there is a `lack of internal knowledge`, addressing the your concern about `positive ratios between splits are similar`.
> ---
> ### Weakness 4: Broader applicability
> Our work's implications extend to a wide range of LLM applications, including generative tasks.
>
> - **Proven effectiveness in a generative task**. Section 5.4 and Fig. 4 **demonstrate that our UU‑trained reward model steers RLHF toward safer outputs**, significantly outperforming a baseline.
> - **Classification underpins modern LLM alignment**. Most training paradigms hinge on binary decisions [2]:
>   - **RLHF/DPO**: Decide which of two responses is preferred. Our pipeline produces these preference labels with minimal human input, even for subjective or domain‑specific traits.
>   - **GRPO**: Relies on feedback labeling answers as correct or incorrect. UU refinement supplies such labels even when the ground truth is unclear.
>   - **Generative guidance**. High-accuracy classifiers, such as those learned via our UU refinement, are essential for guiding generative models toward desired properties (e.g., protein design) [3].
>
> [2] https://dl.acm.org/doi/10.5555/3692070.3694015 \
> [3] https://neurips.cc/virtual/2023/poster/71899
>
> ---
>
> We trust our responses address your concerns. Our method for LLM self-refinement is novel and valuable. Thank you for your time and feedback.

---

> > ### Comment · Reviewer_py1Q · 2025-08-06
> >
> > Thank you for your detailed response. After carefully reviewing your reply, as well as the feedback from other reviewers, I acknowledge that most of my concerns have been addressed.
> >
> > Regarding the point that "Classification underpins modern LLM alignment," I agree that conducting experiments in binary classification do serve as an effective anchor.
> >
> > However, I remain curious about the theoretical guarantees behind the claim that "UU risk remains unbiased as long as the pseudo‑positive split’s true‑positive rate exceeds that of the pseudo‑negative split." Further clarification or theoretically supporting reference on this point would be appreciated.
> >
> > I will update my score accordingly.

---

> ### Author Response · Authors · 2025-08-06
>
> Thank you for your thoughtful reassessment and constructive engagement. We appreciate the chance to further clarify the theoretical foundations of our approach.
>
> ---
>
> The central idea of Unlabeled-Unlabeled (UU) learning is to **construct an unbiased estimator of the supervised classification risk**, $R_{pn}(g)$, such that its expected value accurately represents the true classification risk.
>
> In standard supervised learning, a classifier $g$ is trained by minimizing the risk, calculated from a positive dataset $C_p$ and a negative dataset $C_p$:
> $$R_{pn}(g) = \pi_+ E_{x \sim C_p}[\ell(g(x), +1)] + (1-\pi_+) E_{x \sim C_n}[\ell(g(x), -1)]$$
> where $\pi_+$ is the positive class prior, and $\ell$ is the loss function.
>
> In contrast, **UU learning utilizes two unlabeled datasets**: a pseudo-positive dataset $\tilde{C}_p$ and a pseudo-negative dataset $\tilde{C}_n$, each containing true-positive proportions $\tilde{\theta}_p$ and $\tilde{\theta}_n$, respectively. Note that standard supervised learning corresponds to the special case $\tilde{\theta}_p = 1$ and $\tilde{\theta}_n = 0$.
>
> The UU risk is then defined as:
> $$R_{uu}(g) = a E_{x \sim \tilde{C_p}}[\ell(g(x), +1)] - b E_{x \sim \tilde{C_p}}[\ell(g(x), -1)] - c E_{x \sim \tilde{C_n}}[\ell(g(x), +1)] + d E_{x \sim \tilde{C_n}}[\ell(g(x), -1)]$$
> where $a, b, c, d$ are coefficients that are determined by the $\tilde{\theta}_p$ and $\tilde{\theta}_n$ (see details in `lines 136-137`).
>
> Critically, $R_{uu}$ remains unbiased as an estimator of $R_{pn}$ when $\tilde{\theta}_p > \tilde{\theta}_n$ (i.e., `pseudo‑positive split's true‑positive rate exceeds that of the pseudo‑negative split`), enabling classifier learning without clean labeled data.
>
> This theoretical foundation has been established and empirically validated in existing theoretical research [1, 2]. The theoretical guarantee is provided in Theorem 4 of the original paper [1].
>
> [1] [Lu et al., 2019, "On the Minimal Supervision for Training Any Binary Classifier from Only Unlabeled Data," ICLR](https://openreview.net/forum?id=B1xWcj0qYm)\
> [2] [Lu et al., 2020, "Mitigating Overfitting in Supervised Classification from Two Unlabeled Datasets: A Consistent Risk Correction Approach," AISTATS](https://proceedings.mlr.press/v108/lu20c.html)
>
> ---
>
> Our novel contribution is the **first to conceptualize LLM judgments as weakly supervised signals and to integrate the theoretically grounded UU learning into LLM self-refinement and modern alignment pipelines**. This integration represents a substantial advancement compared to previous methods reliant exclusively on internal LLM knowledge.
>
> ---
>
> **Update for Camera-Ready Version**: We will enhance the Preliminaries section with an expanded discussion of UU learning theory to clearly articulate these theoretical foundations and their practical implications for LLM self-refinement and alignment.

---

> > ### Comment · Reviewer_py1Q · 2025-08-09
> >
> > Thanks for your detailed response regarding theoretical reference. I have no concerns regarding this work. I already change my assessment accordingly.

---

### Official Review · Reviewer_DcaC · 2025-07-06

**Clarity:** 3
**Significance:** 3
**Originality:** 3
**Rating:** 4
**Confidence:** 1

**Summary:**

Overall, the paper is well written, and the problem it aims to solve is very important: how can we improve LLMs without requiring large amounts of expensive labeled data? The key idea of this paper is to leverage two different unlabeled datasets to learn from each other via a known technique called Unlabeled-Unlabeled (UU) learning. The main technical contribution of this paper is to use UU learning to design a new learning pipeline for refining labels. The experiments focus on benchmarks that have high uncertainty or lack high-quality labels.

**Questions:**

From the experiments, most of the benchmarks become stable after only one iteration. Is this expected, or are there any other underlying reasons for this?

I am not sure if there are other approaches to enhance label quality, as the experiments seem to focus only on your newly proposed approach. Could you please clarify this further?

**Ethical Concerns:**

["NO or VERY MINOR ethics concerns only"]

**Limitations:**

yes

**Quality:**

3

**Strengths And Weaknesses:**

Strengths:

 The paper targets a significant problem in my view: improving LLMs without expensive labeled data.
 The experiments show significant improvements on several specific benchmarks, including fake news, safety, protein structure, etc. These benchmarks are not well studied in LLMs and lack high-quality data which make lots of sense to me.
The pipeline of applying UU learning is clear and easy to follow for other researchers

Weaknesses:
The core technical contribution of this paper is the use of UU learning for this new problem. However, since the submission is not in my area, I am unsure about the significance of its technical contribution.
Only one iteration is needed for most of benchmarks with UU learning which is limited the potential of the new method.

---

> ### Author Rebuttal · Authors · 2025-07-31
>
> We appreciate the thoughtful evaluation. Below, we address every point raised.
>
> ---
>
> ### Weakness: Significance of Technical Contribution
>
> Thank you for raising this point. As other reviewers (e.g., `Lj7L`, `e1xr`) have remarked, our approach is `innovative` and `novel in this context`, and **[NeurIPS reviewing guidelines](https://neurips.cc/Conferences/2025/ReviewerGuidelines) explicitly acknowledge** that a `novel combination of existing techniques` is a valuable contribution.
>
> Our novel contributions are:
> - **Novel self‑refinement direction**: We are the first to apply theoretically grounded UU learning to the critical task of LLM self‑refinement, **opening a new avenue for performance gains** where existin methods that rely on an LLM’s internal knowledge have struggled. Figures 2 and 3 demonstrate consistent per‑iteration improvements.
> - **Demonstrated practical utility in LLM post‑training**: We further show effectiveness in LLM post‑training (RLHF) for generative safety alignment (Fig. 4), indicating that our method **offers practical value beyond classification tasks**.
>
> ---
>
> ### Question 1: Performance Stabilizing After One Iteration
>
> Thank you for your clarification.
>
> As you observed, the largest gain occurs in the first iteration because the initial noisy labels allow robust UU to correct the biggest errors. Nevertheless, **Figures 2 and 3 reveal continued performance gains in later rounds**, particularly when the base model is weaker or the dataset is highly noisy (e.g., Protein‑Structure). These incremental improvements confirm that **iterative refinement continues to provide value beyond the initial correction**.
>
> To make this trend unmistakable, we will **add a per‑iteration accuracy table** in the appendix, clearly quantifying the cumulative gains.
>
> ---
>
> ### Question 2: Alternative Approaches to Improve Label Quality
>
> We thank you for this question and apologize if this was not clear in the text.
>
> We did, **in fact, compare our method with strong baselines** described in Sec 5.2 ("Baselines") and Table 1, covering state-of-the-art label‑refinement techniques from related fields:
> - **PIE [65]**: An iterative, noise-robust method from the field of weakly-supervised learning.
> - **CCP [25]**: A method using contrastive learning for label refinement from the field of semi-supervised learning.
> - **Self‑Refinement with GPT‑4o / DeepSeek‑R1**: Uses the LLM’s internal knowledge to iteratively critique, relabel, and improve its outputs.
>
> Our experiments (Figure 2, 3) show that **our proposed pipeline consistently outperforms** these state-of-the-art methods, particularly on more challenging datasets.
>
> ---
>
> We believe these clarifications resolve the your concerns. Given the method’s novelty, empirical strength, and practical relevance, we kindly ask you to consider raising the recommendation.
>
> Thank you again for your careful review.

---

### Official Review · Reviewer_Lj7L · 2025-07-06

**Clarity:** 2
**Significance:** 2
**Originality:** 2
**Rating:** 4
**Confidence:** 4

**Summary:**

The paper presents a label refinement method for improving the classification performance of large language models (LLMs) using a new iterative pseudo-labeling pipeline based on Robust Unlabeled-Unlabeled (UU) learning. The key idea is to leverage two unlabeled datasets with different positive class priors to iteratively denoise LLM-generated pseudo-labels. Unlike traditional self-refinement methods that rely on the LLM's own internal feedback mechanisms, this approach minimizes dependence on internal model knowledge and instead utilizes data-driven refinement through robust UU learning. Experiments across diverse domains—including low-resource languages, patent classifications, protein structure prediction, and LLM alignment—demonstrate performance gains and scalability under minimal human supervision

**Questions:**

See above.

**Ethical Concerns:**

["NO or VERY MINOR ethics concerns only"]

**Limitations:**

The limitations are not discussed. It will be good to discuss the risk of confirmation bias, the requirements of labeled data, and the computational cost of multiple rounds of refinement.

**Quality:**

2

**Strengths And Weaknesses:**

**Strengths**


The main idea to apply robust UU-learning iteratively to refine the pseudolabels of LLMs is interesting and novel in this context.

Overall, the paper is well written with clear flow and presentation of ideas. A few things could be improved in notation. Appreciate the background, brief summary in 4.1, and Figure 1.

Empirical evaluation on several tasks, including sentiment analysis, patents and protein classification, and safety alignment, etc., shows the effectiveness of the proposed iterative method.



**Weaknesses**


The method requires labeled data for estimating priors and other coefficients in the UU loss. Lack of such labeled data may produce unreliable estimates and degraded results.
Do the baselines also use the same amount of labeled data? What if the first round of LLM uses these labeled points as in-context examples?


The same risks of confirmation bias as in pseudolabeling (self-training) still apply in this iterative labeling using UU-learning. There are several pseudolabeling baselines focused on correcting these issues; comparison against those can show the distinctions more clearly.


Presentation (Minor): Notation in section 3.2 can be simplified, $\tilde{C}_p$ and $\tilde{C}_n$ are not actually positive or negative sets (unlike most of the notation in the paper, where subscripts clearly mean positive or negative samples). Maybe $\tilde{C}_1$ and $\tilde{C}_2$ might be good enough to avoid confusion. Colors and markers in the plots are not very clear. Please use easily distinguishable colors and markers and maintain consistency across figures.

---

> ### Author Rebuttal · Authors · 2025-07-31
>
> Thank you for your constructive feedback and for noting the novelty of our approach. We address your points below.
>
> ---
> ### W1:  Requirement for Labeled Data
> Our method indeed uses a small labeled set, but **this requirement is minimal compared with existing techniques**.
>
> - **Tiny supervision, large gains**. With just 50–100 labels, our model achieve iterative performance gain (Figures 2–3), whereas strong baselines plateau, or even decline, with the same supervision. Conventional fully‑supervised pipelines need thousands of labels; our method therefore slashes annotation cost by 10–1000×.
> - **Value in expert‑scarce domains**. Patent and protein tasks, where annotation is costly, still benefit markedly, demonstrating practicality in low‑resource settings.
> - **Why other methods fail**. Prior self-refinement approaches cannot transform additional labels into meaningful signals. Table A shows accuracy stays flat or even drops even when adding 100 in-context examples with LLM's self-refinement. Robust UU, by contrast, does leverage those labels effectively.
>
> ---
>
> ### W2  Confirmation‑Bias Risk
> We share your concern and **purposely design our method to mitigate it**:
> - **External, data‑driven correction**. The LLM’s potentially biased knowledge is used only once to generate the initial pseudo‑labels. From iteration 1, learning is purely data‑driven: a discriminative classifier trained with robust UU risk gradually removes residual noise in those initial labels.
> - **Empirical evidence**. Accuracy rises monotonically across all datasets (Figures 2–3), even when initial pseudo‑labels are noisy (e.g., ≤ 60 % on Protein).
> - **Planned mitigations**. We believe bias-reduction techniques can be integrated into our method and would welcome relevant literature to guide our camera-ready revision and future work.
>
> ---
>
> ### Questions
> > Do the baselines also use the same amount of labeled data?
>
> We conducted a fair comparison about the supervision as follows.
>
> - **CCP (Semi-Supervised)**: Employs the same 100 labeled examples as our method used, following the CCP protocol.
> - **PIE (Weakly-Supervised)**: We add the same 100 labeled examples to PIE’s training data, alongside original pseudo-labels.
> - **LLM Self-Refinement Frameworks**: These rely solely on in‑context learning. Because longer input often degrade accuracy [1,2], we used 10 few‑shot examples in the main runs. To answer your question, we re-ran with 100 examples using gpt-4o-mini; as Table A shows, accuracy is nearly unchanged from the 10-example case (see Figure 3) and flat accuracy across iterations.
>
> Table A: Self‑Refinement Accuracy (GPT‑4o‑mini, 100 in‑context examples)
>
> | dataset | Iter0 | Iter1 | Iter2 | Iter3 | Iter4 | Iter5 |
> |---------|-------|-------|-------|-------|-------|-------|
> | corona  | 0.734 | 0.738 | 0.745 | 0.727 | 0.730 | 0.738 |
> | patent  | 0.698 | 0.696 | 0.695 | 0.691 | 0.691 | 0.683 |
> | protein | 0.630 | 0.594 | 0.606 | 0.626 | 0.617 | 0.620 |
>
> This setup provides **each baseline with consistent supervision**, matching its paradigm and ensuring a fair comparison. Also, Table A indicates that **existing self‑refinement struggles to extract useful signals** from additional labels, while our UU‑based approach overcomes this limitation.
>
> [1] https://aclanthology.org/2025.acl-long.1396/\
> [2] https://aclanthology.org/2023.findings-emnlp.745/
>
> ---
>
> ### Notation and Clarity of Figure
> We are grateful for these concrete suggestions for improving the paper's presentation.
>
> - **Notation**: We will revise the notation for pseudo-positive and pseudo-negative corpora in the camera-ready version to avoid confusion, as suggested.
> - **Figures**: Plots will be revised for the camera-ready version to use more distinguishable, consistent colors and markers across all figures. For clarity, we'll add **tables with the exact numerical data from each figure**.
>
> ---
>
> We appreciate your thoughtful review and valuable feedback. We look forward to addressing your concerns in our camera-ready submission.

---

### Comment · Area_Chair_SkyQ · 2025-08-05
**Author-Reviewer Discussion Phase Closing Soon**

Dear Reviewers,

Thank you for your support of NeurIPS 2025!

Please be reminded that the Author-Reviewer Discussion phase is nearing its end (August 6, 11:59pm AoE). If you have not yet responded to the author’s rebuttal or provided any feedback, kindly do so before the window closes.

Best regards,

AC

---

### Note · Authors · 2025-08-13

We deeply appreciate the constructive discussion with the AC and reviewers.

---

**Contribution and impact**: Our paper studies LLM self‑refinement when the model’s internal knowledge is limited and existing methods relying on it have struggled. We replace model‑internal critique with a data‑centric pipeline that iteratively denoises pseudo‑labels via robust UU learning. Reviewers characterized our method as `novel in this context`, `targeting a significant problem`, and `innovative`, and they noted strong empirical results across diverse benchmarks, including benefits for RLHF safety alignment.

---

During discussion, concerns were clarified as follows.

**Binary scope**: Our experiment focused on binary tasks as a clear anchor for studying self‑refinement. Reviewers recognized a path to multi‑class (e.g., complementary‑label reductions) and considered the **concern resolved**; we will outline it in the camera‑ready.

**Noise‑reduction dynamics (e.g., Confirmation bias & Gain after One‑iteration)**: After initial LLM labeling, learning switches to a classifier trained with robust UU risk, using data‑driven features rather than the LLM’s internal knowledge. This design targets bias in the initial labels: the first iteration yields the largest correction, with further gains in later rounds on most datasets; we will pair each figure with a per‑iteration accuracy table to make the trend clear.

**Use and fairness of labeled data**: With as few as 50 labels, our method achieves clear iterative gains that existing baselines do not achieve. Supervision are set to be fair across methods. With the same labels, baselines plateau or degrade, while our UU pipeline continues to improve. The camera‑ready version will make experimental protocols explicit.

**Failure case study (Fake‑News/Gemma)**: We will include a case study analyzing this single anomaly (per‑sample difficulty profiling, TP/FP/FN/TN decomposition, and embedding‑space error structure) and will release the initial pseudo‑labels to enable replication and further analysis.

**Presentation**: We will refine notation, expand related work, enhance figures, and include precise tables.

---

In summary, reviewers judged the problem significant, the direction novel, and the evidence strong, including RLHF benefits. Most concerns were resolved in discussion; we will further strengthen the paper for the camera‑ready. Given this recognition, we believe that sharing these findings with the community is valuable.

---

### Decision · Program_Chairs · 2025-09-17

**Decision:**

Accept (poster)

**Comment:**

The paper introduces a data-centric self-refinement pipeline that iteratively denoises LLM pseudo-labels via robust Unlabeled–Unlabeled (UU) learning. It shows consistent gains across diverse binary tasks and demonstrates utility for RLHF safety alignment.

As pointed by reviewers, the approach is clearly presented and shows consistent gains across several benchmarks, with additional benefits to RLHF through improved reward models. During discussion, key reservations were addressed. The authors clarified UU risk conditions and theory, fairness of supervision across baselines, and provided plans to add per-iteration tables, failure-case analysis, and improved figures. Overall, the work advances self-refinement beyond model-internal critique and provides a principled mechanism that is practical in label-scarce regimes and impactful for alignment.